# In-section Click-iT detection and super-resolution CLEM analysis of nucleolar ultrastructure and replication in plants

Michal Franek [1] ✉, Lenka Koptašíková [2,3], Jiří Mikšátko[2], David Liebl[2], Eliška Macíčková [2], Jakub Pospíšil [4], Milan Esner [4], Martina Dvořáčková [1] ✉ & Jiří Fajkus [1,5]

Correlative light and electron microscopy (CLEM) is an important tool for the localisation of target molecule(s) and their spatial correlation with the ultra-structural map of subcellular features at the nanometre scale. Adoption of these advanced imaging methods has been limited in plant biology, due to challenges with plant tissue permeability, fluorescence labelling efficiency, indexing of features of interest throughout the complex 3D volume and their re-localization on micrographs of ultrathin cross-sections. Here, we demonstrate an imaging approach based on tissue processing and embedding into methacrylate resin followed by imaging of sections by both, single-molecule localization microscopy and transmission electron microscopy using consecutive CLEM and same-section CLEM correlative workflow. Importantly, we demonstrate that the use of a particular type of embedding resin is not only compatible with single-molecule localization microscopy but shows improvements in the fluorophore blinking behavior relative to the whole-mount approaches. Here, we use a commercially available Click-iT ethynyl-deoxyuridine cell proliferation kit to visualize the DNA replication sites of wild-type *Arabidopsis thaliana* seedlings, as well as *fasciata1* and *nucleolin1* plants and apply our in-section CLEM imaging workflow for the analysis of S-phase progression and nucleolar organization in mutant plants with aberrant nucleolar phenotypes.

Correlative light and electron microscopy (CLEM) combines the unique benefits of light and electron microscopy (EM), where images of specific, fluorescently labelled cellular structures captured in the fluorescence microscope are spatially correlated with a morphological map of ultrastructural features revealed by electron microscopy. In practice, correlative microscopy approaches often entail a number of challenges, namely in the choice of an appropriate chemical fixation, contrasting, resin embedding and labelling steps that have to suit both, preservation of fluorescence (or antigens for immunofluorescence staining) and preservation of fine ultrastructure for electron

[1]Mendel Centre for Plant Genomics and Proteomics, Central European Institute of Technology (CEITEC), Masaryk University, Kamenice 5, CZ-62500 Brno, Czech Republic. [2]Charles University, Faculty of Science, Biology Section, Imaging Methods Core Facility at BIOCEV, Průmyslová 595, 252 50 Vestec, Czech Republic. [3]University of Exeter, Faculty of Health and Life Sciences, Bioimaging Centre, Geoffrey Pope Building, Stocker Road, EX4 4QD Exeter, UK. [4]Cellular Imaging Core Facility CELLIM, Mendel Centre for Plant Genomics and Proteomics, Central European Institute of Technology Masaryk University (CEITEC MU), Kamenice 5, CZ-62500 Brno, Czech Republic. [5]Laboratory of Functional Genomics and Proteomics, National Centre for Biomolecular Research, Faculty of Science, Masaryk University, Kamenice 5, CZ-61137 Brno, Czech Republic. ✉e-mail: 357550@mail.muni.cz; dvorackova.martina@gmail.com

microscopy. Conventional chemical fixation, post-contrasting with osmium and dehydration prior to sample embedding in resin for transmission EM often results in partial extraction or denaturation of proteins, loss of fluorescence (in the case of fluorescent proteins) or leads to an inefficient post-embedding immunolabelling due to the destruction or limited accessibility of epitopes on the surface of resin sections. The last decade brought forward advances in sample preparation for CLEM, which demonstrated the feasibility of ultrastructural analysis by EM while maintaining the fluorescence of endogenous proteins with the possibility of super-resolution imaging[1–3] or protocols that enable post-embedding on-section fluorescence labelling[4]. There has been considerable progress in endogenous labelling strategies, with different groups successfully developing SMLM-compatible, fixation-resistant fluorescent proteins, such as frSkylan_S[5] or mEos4b[6]. While the introduction of these constructs for stable expression in plants is time-consuming, there is an alternative for investigating structures in plant cells using exogenous labelling with low-molecular-weight compounds. The combination of super-resolution microscopy (SRM), especially single-molecule localization microscopy (SMLM), with the EM brings about further technical considerations. SMLM uses the stochastic photo-switching ("blinking") of fluorophores to determine their positions with resolution below the diffraction limit (~5–40 nm), therefore retention of the blinking potential of fluorophores upon embedding in a resin as well as epitope accessibility and reactivity in resin sections are crucial. To date, few groups have reported successful implementations of SRM-CLEM[7,8], mostly on adherent mammalian cell lines[2,3,9]. Implementation of these techniques in plant experimental models is not trivial, since plant cell walls represent a considerable obstacle for the efficacy of the highly-specific labelling of intracellular structures required for localization microscopy[10,11].

Novel methodological approaches in super-resolution microscopy/nanoscopy are generally benchmarked on well-described cellular structures, such as the nuclear pore complex or cytoskeletal components[12,13]. Here, we developed a correlative workflow to study nucleolar architecture changes during progression of DNA replication in plant cells. We have previously characterized the organization and replication of ribosomal DNA in plant nuclei[14,15] using fluorescence in-situ hybridization and replication labelling. We have shown that the replication of the 45S ribosomal DNA (rDNA) occurs throughout S-phase in the nucleus and nucleolus, whereby the active intranucleolar fraction of rDNA replicates in early-S-phase, larger intranucleolar clusters of rDNA genes replicate in the mid-S-phase and the perinucleolar, transcriptionally inactive fraction replicates in the late-S-phase. Results from replication labelling combined with sequencing (Repli-SEQ) have shown that nucleolus displays a bipartite replication pattern, with distinct segments of rDNA replicating in the early and in the late-S-phase, in agreement with microscopic observations[16,17].

To investigate the links between nucleolar architecture and replication progression in the nucleolus, we selected specific *A. thaliana* mutants, known for alterations in nucleolar structure (*nucleolin1, nuc1*) or progressive loss of ribosomal genes (*fasciata1, fas1*). It has been previously reported that *nuc1* mutants display changes in nucleolar architecture[18–20], with changes in the methylation pattern of rDNA genes. *Fas1* mutants have been shown to have a decreased number of rDNA copies and exhibit general chromatin decompaction[21,22].

In this work, we establish a workflow for consecutive-section and same-section CLEM (nomenclature from ref. [6]) in the tissue sections of the *A. thaliana*. In this workflow, the molecules of interest are rendered fluorescent through the utilization of Click-iT chemistry—a technique awarded Nobel Prize in Chemistry in 2022—which circumvents limited penetration of antibodies and enables fluorescence labelling throughout the entire volume of the tissue section. Furthermore, this approach mitigates the typical trade-off between preserving ultrastructure and immunogenicity since it allows the use of higher

concentrations of glutaraldehyde (up to 3%) to cross-link the tissue before target molecules are subjected to fluorescence labelling.

In this study, we integrate the analysis of replication with SMLM and of nucleolar structure by transmission electron microscopy (TEM), with the localization precision in the range of 10–20 nm for SMLM data. We demonstrate that (i) imaging of plant tissue segments embedded in Lowicryl is feasible by both, SMLM and TEM; (ii) SMLM performed on 500 nm thick sections has high-quality metrics, (iii) resin sections are permeable for low molecular weight (MW) labelling compounds (Click-iT chemistry, or e.g. phalloidin labelling) and allow for labelling throughout the section volume and (iv) same-section CLEM on 150 nm sections yields sufficient fluorescence signal for confocal imaging and allows precise correlation of fluorescence with ultrastructural features in TEM micrographs. We demonstrate that rDNA replication patterns in plants can be precisely localised and visualized by (pre-)labelling of samples with ethynyl-5'-deoxyuridine (EdU) followed by chemical processing, resin embedding and performing Click-iT chemistry detection directly on semithin sections. While serial sectioning of alternating semithin (500 nm) and ultrathin sections (70 nm) allows us to probe the ultrastructure of the nucleolus with TEM and the localization of fluorescently labelled replication patterns in the adjacent section by SMLM, the same-section CLEM approach in this workflow offers precise correlation of structures visible in the two imaging modalities. Our data suggest that the presence of intranucleolar replication foci (fluorescent labelling) is concomitant with the detection of fibrillar centres in the nucleoli (discernible on TEM micrographs) and the differences in the nucleolar architecture correlate with changes in the size of intranucleolar replication foci between wt, *fas1* and *nuc1* plants.

## Results
### Correlative workflow for large plant tissue segments

To access the ultrastructure of the plant nucleolus in relation to the S-phase progression, we performed Click-iT chemistry labelling on semithin sections of plant roots embedded in EM resin. First, we incubated seedlings with EdU, fixed the samples by chemical cross-linking and embedded samples into low-melting agarose blocks for easier handling of the roots (stem and leaves trimmed off) during subsequent steps of chemical processing, as reported in ref. [23] (Fig. 1A, B). Next, two types of resin were tested for sample embedding and *in-section* Click-iT labelling: (i) Spurr's resin, an epoxy-based, high-penetration, low-viscosity, hydrophobic resin and (ii) Lowicryl K4M, a methacrylate-based, water-compatible, polar resin (referred to further in the text as "Lowicryl sections" for simplicity; Fig. 1C, D). The Click-iT replication labelling was then performed on 500 nm and 150 nm longitudinal sections prepared from the resin-embedded seedlings (Fig. 1E, G), while the uranyl acetate and lead citrate post-contrasting was performed on alternate 70 nm sections for consecutive-section CLEM and 150 nm sections for same-section CLEM (Fig. 1F). The highest signal-to-noise ratio was achieved in Lowicryl-embedded samples (Fig. 2A), with discernible differences in EdU patterns reflecting the different stages of DNA replication (Fig. 2E–H). We found that embedding of samples into Spurr's resin, which includes conventional osmium tetroxide contrasting, is also compatible with Click-iT labelling on resin sections, although it produces considerable background fluorescence (Fig. 2C, D).

Given that different fixation and embedding conditions might affect the ultrastructural preservation of the sample, we looked at the ultrastructure of nuclei, organelles as well as plant cell connections—plasmodesmata (Supplementary Fig. 1, Supplementary Data 1). As expected, the Spurr's protocol produced better ultrastructural preservation of the sample, altogether with a better contrast for organelles (Supplementary Fig. 1A, B and details in C–F). Lowicryl embedding and milder fixation conditions still preserve sufficient ultrastructural detail for the analysis of nucleolar architecture (Supplementary Fig. 1G, H) as

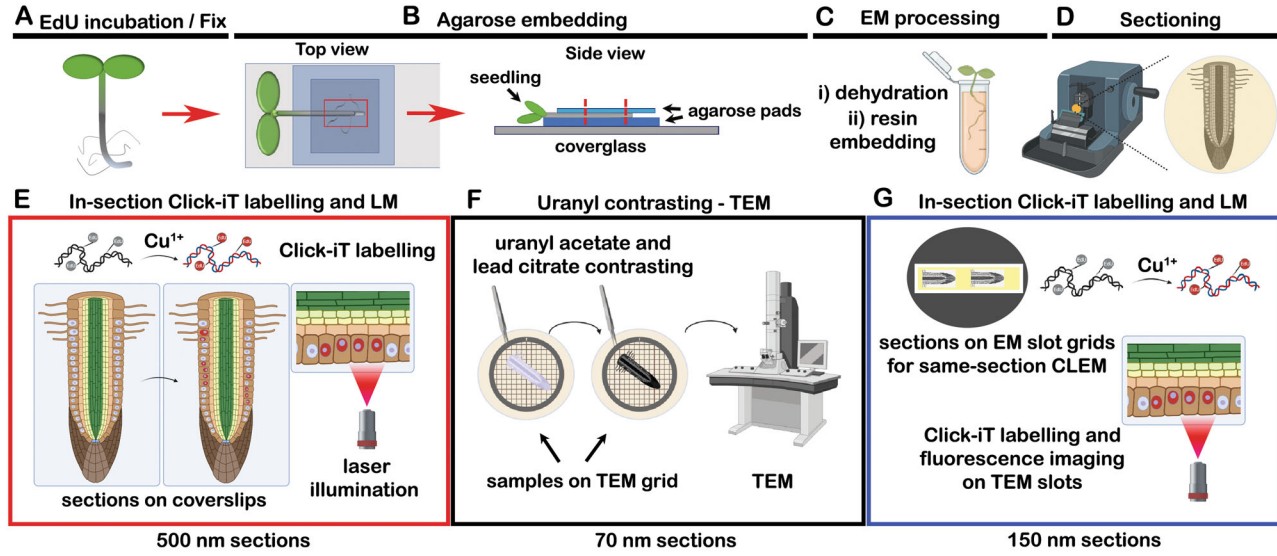

**Fig. 1 | Schematic representation of the CLEM workflow for *A. thaliana*.** Seedlings were labelled with EdU, fixed in a mixture of glutaraldehyde and formaldehyde (**A**) and embedded into a block of low-melting agarose (**B**). Strips of agarose-embedded roots were post-fixed, dehydrated and embedded into Spurr's or Lowicryl resin (**C**). Polymerized resin blocks were sectioned (**D**) with an ultramicrotome into consecutive pairs of semithin (500 nm) sections for fluorescence microscopy and ultrathin sections (70 nm) for electron microscopy, as well as 150 nm sections for same-section CLEM. Click-iT labelling was first performed on 500 nm sections (**E**) and images were captured by either spinning disk confocal microscopy or SMLM. Consecutive ultrathin sections were then collected on EM slot grids, post-contrasted with 4% uranyl acetate and 3% Reynold's lead citrate, and imaged in a TEM (**F**). For same-section CLEM, the samples were stained by Click-iT labelling and imaged on EM slot grids by spinning-disk confocal microscopy, with subsequent post-contrasting prior to TEM (**G**). LM light microscopy, TEM transmission electron microscopy.

well as recognition of different plant organelles (Supplementary Fig. 1I–L), including the internal structure of mitochondria (Supplementary Fig. 1, insets I–L). Crucially, the Lowicryl embedding protocol presented here omits the osmium tetroxide fixation/contrasting and offers superior properties for super-resolution microscopy (as discussed below). A gallery of TEM images for the comparison of ultrastructure preservation between Spurr and Lowicryl protocols in different plant mutants can be found in the Supplementary Data 1.

Having established that embedding of samples in selected resins does not preclude Click-iT detection, we aimed to analyse whether image registration is possible between fluorescence and EM micrographs capturing large segments of plant tissue sections (Fig. 3A, B). In general, image registration for CLEM is accomplished either by mapping defined cellular structures (e.g. labelled mitochondria; ref. 8) or using fluorescently labelled gold nanoparticles as fiducial markers[1,7]. To align the images from electron microscopy and light microscopy, we took advantage of the anatomical features of the root and cell wall morphology, visible in both modalities (Fig. 3C, D; Fig. Supplementary Fig. 2) and aligned these using the ec_CLEM software package[24]. Consistent with previously published literature[1,8], we noted shrinking, stretching and distortion artifacts between sections (Fig. 3E, F). Ultimately, we were able to correlate replication patterns obtained from fluorescence microscopy with the corresponding TEM micrographs locally for individual cells (Fig. 3G–I). The percentage of cells labelled with Click-iT chemistry progressing through S-phase during the 90 min incubation with EdU was 26.1% (*n* = 330 cells). We found that for 65.2% of cells labelled with Click-iT chemistry we found a well-defined nucleus in the corresponding EM micrographs (*n* = 46 cells), suggesting approximately a third of the cells cannot be correlated due to the offset in the axial plane.

### Lowicryl sections are compatible with 2D and 3D single-molecule localization microscopy
Having demonstrated the possibility of Click-iT labelling on Lowicryl sections in a correlative workflow, we next focused on mapping the

ultrastructural details of replicating chromatin segments by implementing SMLM in cells tagged with Click-iT chemistry. We reasoned that the embedding of samples into Lowicryl and sectioning would reduce sample complexity (e.g. minimal thickness, sample immobilization) and improve SMLM characteristics. Click-iT labelling in Lowicryl sections, resulted in sufficient photo-switching properties, averaging 128,780 localizations (*n* = 28 images, s.d. = 61,275; with the number of localizations normalized for the number of cells in the field of view) with minimal out-of-focus detections given the physical size of the specimen (500 nm). Besides localization microscopy metrics such as localization precision or the average number of localizations, we used Nano-J SQUIRREL to evaluate possible errors in image reconstruction (Supplementary Fig. 3). Overall, we observed a significant correlation between widefield and super-resolution images, with global resolution-scaled Pearson correlation coefficients for analysed images (RSP) being 0.96, 0.91 and 0.76 (Supplementary Fig. 3C, F, I). The image resolution, calculated from the Fourier ring correlation analysis, was approximately 18 nm (*n* = 10; SD = 8.8 nm; Supplementary Fig. 3J–M), within the expected range (10–40 nm).

We hypothesized that the improved imaging characteristics for replication labelling on sections could be due to both reduction of sample complexity as well as good accessibility of the Alexa Fluor 647-azide conjugate to the epitopes embedded in the resin. To visualize the depth of penetration for the Alexa Fluor 647-azide conjugate into the resin we prepared and imaged 1000 nm (Fig. 4A, B) as well as 500 nm (Fig. 4C, D) thick sections of the root tissue. As seen in Fig. 4, we detected localizations from the entire volume of the section, visible in the XZ projections (Fig. 4A, C) and 3D projections (Fig. 4B, D). The number of localizations for 3D reconstructions is considerably lower than for 2D (35,717 average localizations, *n* = 9 images, s.d. = 15,234, as opposed to 128,780 average localizations in 2D), probably due to the filtering of a subset of localizations (aberrant PSF deformation).

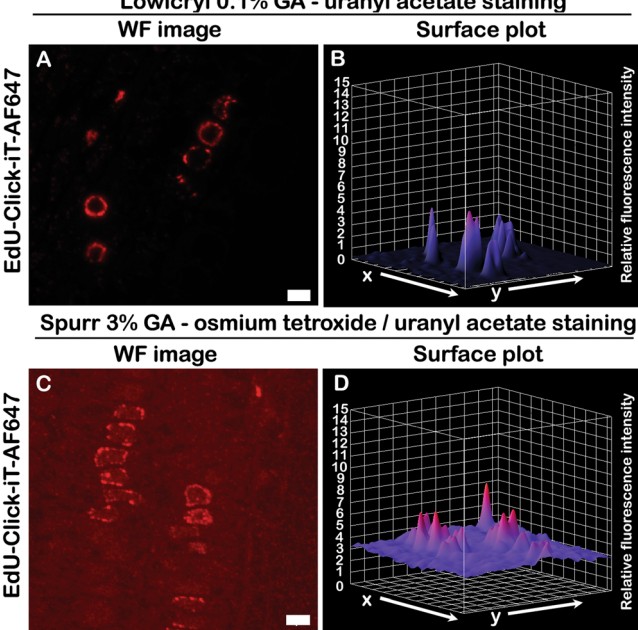

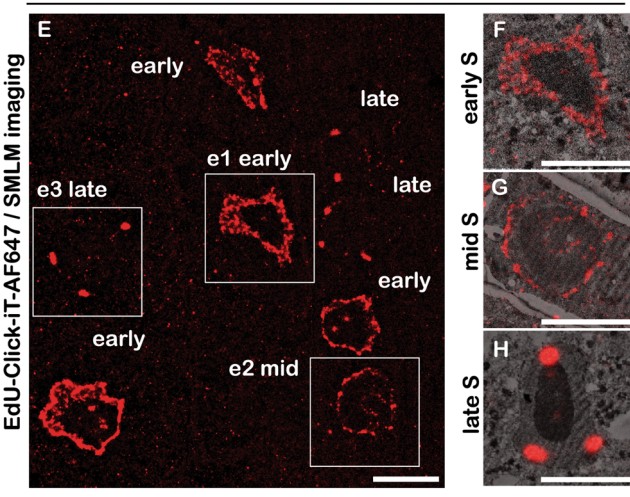

**Fig. 2 | Replication labelling with EdU and Click-iT chemistry in Spurr's and Lowicryl embedded samples.** Plant root sections embedded in Lowicryl (**A**) and Spurr's (**C**) resin labelled with Click-iT chemistry, with surface plots showing the relative background levels in the respective fixation and embedding conditions (**B**, **D**). Large field of view reconstruction of the SMLM image highlighting cells in different stages of DNA replication (**E**). Nuclei showing distinct replication patterns (**F**−early-S-phase, **G**−mid-S-phase and **H**−late-S-phase) are displayed in an overlay with the corresponding TEM micrographs. Spurr's and Lowicryl embeddings were performed in two biological replicates (BR) and two technical replicates (per BR). Scale bar: **A**, **C**: 50 μm, **E** (+insets): 5 μm.

## Immunolabelling of histone H3 and phalloidin tagging of actin filaments on Lowicryl sections

After confirming that EdU-labelling coupled with Click-iT chemistry is efficient for tagging DNA (Figs. 2 and 3A), and for imaging by SMLM (Fig. 4A−D, Supplementary Fig. 3), we tested whether other labelling strategies can be used for Lowicryl sections. We found that dual labelling of chromatin with histone H3 by indirect immunofluorescence and DNA replication sites by Click-iT chemistry also yields sufficient levels of signal when applied on sections and can be used to demonstrate the proportion of EdU-tagged replicating cells in the cell population (Fig. 5A). Notably, the labelling of chromatin with an anti-H3 antibody was restricted to the section surface (Fig. 5B−D) with poor performance

in SMLM in comparison to Click-iT labelling in terms of signal-to-noise ratio and the number of localizations (Supplementary Fig. 4A, B). The discrepancy in performance between H3 and EdU detection was likely due to several factors, including the density of target epitopes, fluorophore properties and limits in resin permeability for the relatively large IgG antibody (14 nm). We next probed the labelling of cellular structures with the low MW compound phalloidin conjugated to Alexa Fluor 647, which labels actin filaments in root sections. As seen in Supplementary Fig. 5, we detected strong labelling adjacent to cell walls with filaments extending into the cytoplasm (Supplementary Fig. 5−Insets), suggesting that detecting cellular structures with low MW compounds is feasible for Lowicryl resin-embedded tissue sections. Overall, we have demonstrated that Lowicryl embedding of samples is compatible with SMLM and allows for efficient labelling with Click-iT chemistry, low MW compounds or immunolabelling.

## Alterations in the nucleolar architecture and intranucleolar replication foci size in the *fas1* and *nuc1 Arabidopsis thaliana* mutants

The protocol we had established was then used to study the links between intranucleolar replication and changes in nucleolar architecture. Replication labelling allows for the discrimination of the S-phase progression, altogether with the identification of intranucleolar replication foci (IRFs). In terms of S-phase progression, cells displaying a diffuse signal in the nucleoplasm were identified as early-S-phase (Fig. 6A−C; E−G, illustrated in D, H), cells with partial clustering and strong signal in the nuclear periphery as mid-S-phase (Fig. 6I−K, illustrated in L), and cells with only a few large chromocenters as late-S-phase (Fig. 6M−O, illustrated in P). In combination with TEM, we were able to link the distribution of IRFs with the architecture of the nucleolus, especially the presence and size of fibrillar centres (FCs), not visible in light fluorescence microscopy. From the pooled data (wt, *fas1* and *nuc1*) of cells in the early-S and mid-S-phase (Fig. 6A−I), nuclei without prominent IRFs or FCs or containing both structures (Fig. 6S; 54% and 32%, respectively, $n = 28$) were found most frequently. Cells in the late-S-phase displayed FCs (Fig. 6M), but did not show IRFs (Fig. 6N, 88.8%; $n = 9$). The quantification (Fig. 6R, S) revealed that the presence of FCs together with IRFs is specific to early- and mid-S-phase.

To test whether the size of IRFs and FCs differs between wt and *fas1* or *nuc1* mutant plants, we calculated the area of IRFs from SMLM reconstructions (Fig. 7A, B) and FCs from TEM data (Fig. 7D, E). Mapping of IRFs from SMLM reconstructions revealed the average area of IRFs to be 129.8 (n - IRFs = 29; n - nuclei = 11; s.d. = 54.8), 165.9 (n - IRFs = 26; n - nuclei = 13; s.d. = 48.8) and 176.7 (n - IRFs = 18; n - nuclei = 13; s.d. = 39.3) nm for wt, *fas1* and *nuc1* plants, respectively (Fig. 7C). For FCs, we found an average area of 293.1 (n - FCs = 31; n - nuclei = 14; s.d. = 116.2), 183.5 (n - FCs = 54; n - nuclei = 15; s.d. = 106.4) and 217.7 nm (n - FCs = 39; n - nuclei = 10; s.d. = 117.1) in the wt, *fas1* and *nuc1* plants, respectively (Fig. 7F). The average size of IRFs in *fas1* and *nuc1* mutants was significantly higher relative to IRFs in the wt (Kruskal−Wallis H-test, $p = 0.0021$, H-statistic = 12.3; Wilcoxon rank-sum test adjusted $p$-value = 0.0072 between wt and *fas1*; and $p = 0.0016$ between wt and *nuc1*). Likewise, mutant plants had a reduced size of FCs (Kruskal−Wallis H-test; $p = 5.83 \times 10^{-6}$; H-statistic = 24.1; Wilcoxon rank-sum test adjusted $p$-value = $9.9 \times 10^{-7}$ between wt and *fas1*; $p = 0.0025$ between wt and *nuc1*). The average number of structures per nucleus also differs between wt, *fas1* and *nuc1* plants. We observed that the average number of FCs per nucleolus in the wild-type was 2.21, whereas it is higher in *fas1* and *nuc1* plants, respectively (4.0 and 3.8 FCs per nucleus for *fas1* and *nuc1*; statistically significant difference between wt and *fas1* plants, $p = 0.0033$). We did not observe statistically significant changes in the number of IRFs in the nucleolus between wt and mutant plants (wt = 2.23; *fas1* = 2.08; *nuc1* = 1.38).

Besides changes in the size of the FCs and IRFs, *nuc1* mutants showed altered nucleolar morphology, in line with previous reports[20].

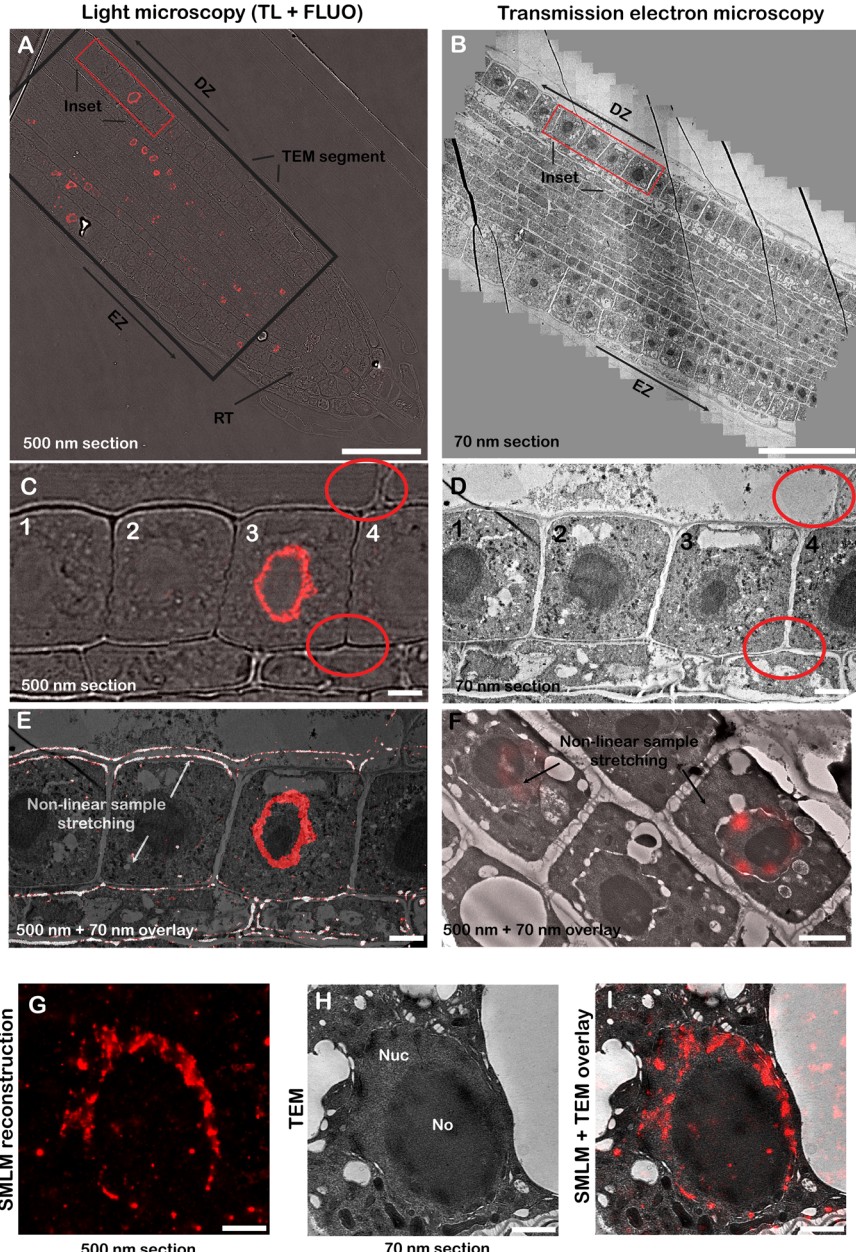

**Fig. 3 | Correlative light and electron imaging of *A. thaliana* roots.** Spinning-disk fluorescence (**A**) and transmission electron microscopy (**B**) imaging of roots on 500 nm and 70 nm sections, respectively. Composite (panorama) image of individual EM fields of view to capture an overview of a large area of the root (**B**). Insets, highlighted by red rectangles in the panels (**A**) and (**B**), magnified in (**C**) and (**D**), showcasing cells of interest (1–4) in the differentiation zone of the root, with anatomical features of the root used for cell identification highlighted by red circles. Anisotropic sample stretching is shown in an overlay CLEM image (**E**, confocal image thresholded to highlight cell walls). Arrows highlight the problems with image registration, shown on cell wall architecture (notice the mismatch of cell wall contours at the bottom and good fitting at top of the image). Anisotropic stretching shown in the displacement of the fluorescence signal from nuclei (**F**), not fitting the underlying cell localization observed in TEM data. Detailed view of a root nucleus shown in SMLM reconstruction (**G**). Details of the same cell in the corresponding EM image (**H**) and image overlay shown in (**I**). TL transmitted light, FLUO fluorescence, EZ elongation zone, DZ differentiation zone, RT root tip, Nuc nucleus, No nucleolus, IRF intranucleolar replication focus. Consecutive-section CLEM was perfomed in five biological replicates. Scale bar (**A**, **B**): 50 μm, (**C**–**E**): 5 μm, (**F**–**I**): 2 μm.

The phenotypic changes such as less pronounced FCs with an atypical shape and weaker contrast in the nucleoli were detected in sections from both Spurr's and Lowicryl-embedded samples, implying these occur irrespective of fixation (Supplementary Fig. 6A, C and Supplementary Data 1) and embedding (Supplementary Fig. 6B, D and Supplementary Data 1) conditions. Altogether, we found significant differences between wild-type and mutant samples in the replication of nucleolar material and corresponding changes in the nucleolar ultrastructure.

## Same-section CLEM on 150 nm sections shows colocalization of IRFs and FCs in plant nucleoli
While the quantification of the size of intranucleolar structures such as the IRFs and FCs benefits from consecutive-section CLEM and super-resolution approaches, the correlation of the mutual position of the FCs and IRFs lacks precision due to the axial offset between consecutive sections. Based on the results of consecutive-section CLEM suggesting the partial overlap of the IRFs and FCs (Fig. 6E–G), we implemented a same-section CLEM approach on 150 nm sections to

## Click-iT labelling on sections

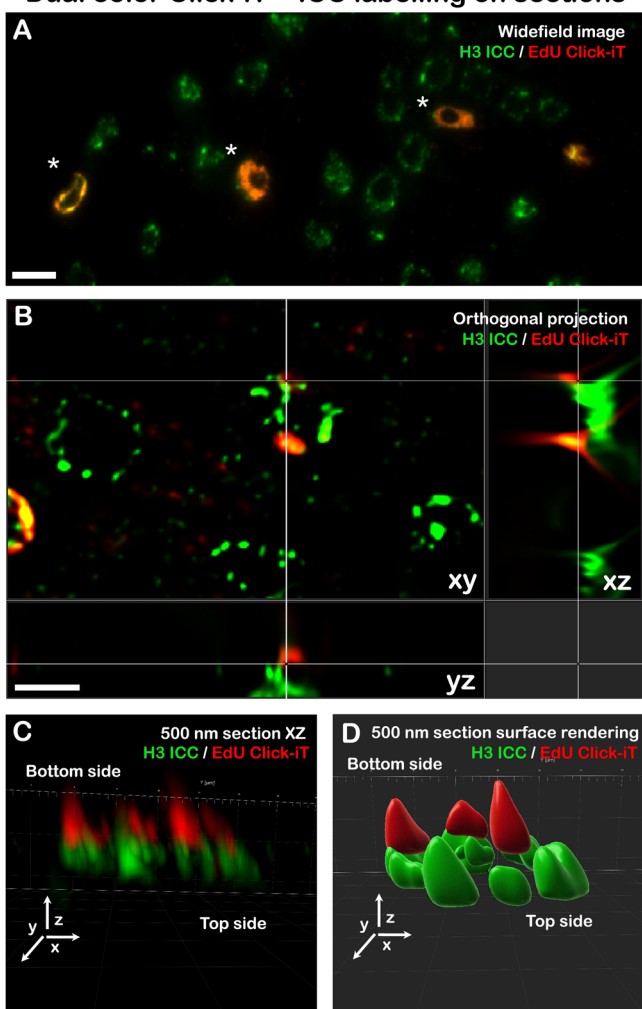

**Fig. 4 | 3D single-molecule localization microscopy on Lowicryl sections.**
Comparison of 1000 nm (**A**, **B**) and 500 nm (**C**, **D**) section labelling with Click-iT chemistry shown in the XZ projection (**A**, **C**) and 3D rendering (**B**, **D**). 3D single-molecule localization microscopy on both 1000 nm and 500 nm sections was performed in two technical replicates. Scale bar: 1 μm.

correlate the distribution of these structures in the nucleolus with higher precision. As a proof of concept, we imaged the *fas1* mutants, as they display larger IRFs which are then more likely to be detected in sections of 150 nm thickness. These sections, collected on EM slot grids were first imaged by confocal fluorescence microscopy and subsequently processed for post-contrasting and TEM imaging. We found that the fluorescence intensity retrieved from these sections was still sufficient to identify distinct stages of nucleolar replication, suggesting that at least for some structures and fluorescence labelling intensities, the same-section CLEM (using 150 nm sections) and consecutive CLEM (using 500 nm sections) are equally applicable. However, in contrast to 500 nm/1000 nm Lowicryl sections, detection of the IRFs in the 150 nm sections was more challenging due to the limited number of intranucleolar signals and higher fluorescence background. Using 150 nm sections, we identified cells with the FCs in the nucleolus (Fig. 8A–C) that did not display the IRFs (Fig. 8D) as well as several cells showing the IRFs correlating with the FCs on TEM micrographs (Fig. 8E–F; H, I and line profiles in K, L). Out of 10 sections analysed, we identified 42 nuclei in S-phase, out of which 7 displayed the IRFs. 4 nucleoli displayed colocalization with the FCs (~57%), suggesting that the replication of a subset of intranucleolar rDNA occurs inside of the FCs.

### Analysis of replication progression in wild-type, *fas1* and *nuc1* mutant plants using SMLM imaging

Replication of the nuclear genetic material is a multi-step process and involves chromatin remodelling, which is required for DNA polymerase access to the replisome complexes. To investigate whether histone chaperone mutants such as *fas1* or *nuc1* exhibit changes in replication progression with manifestation in the nuclear replication pattern, we conducted density-based clustering analysis of SMLM data acquired on semithin sections. First, we performed SMLM image reconstruction in SMAP (Fig. 9A), then filtered the localizations by clustering density (Fig. 9B) and performed density-based spatial clustering of applications with noise (DBSCAN) (Fig. 9C). We chose DBSCAN clustering, as it is a robust technique that does not require specification of the number of clusters, and is not sensitive to cluster shape[25]. In all conditions, we observed a spectrum of replication patterns, ranging from dozens up to hundreds of clusters per nucleus observed after thresholding. We found that differences between replication foci clustering in wt, *fas1* or *nuc1* plants were statistically insignificant (Kruskal–Wallis *p*-value = 0.078) (Fig. 9D).

## Dual color Click-iT + ICC labelling on sections

**Fig. 5 | Click-iT chemistry replication labelling and histone H3 immunolabelling.** Dual labelling of histone H3 and DNA replication by EdU on 500 nm sections (**A**), asterisks (*) highlight replicating cells labelled with EdU-AlexaFluor 647, widefield microscopy. Different depth penetration of the antibody and Click-iT labelling are shown in the orthogonal projection (**B**), with detailed views in (**C**) and surface rendering shown in (**D**). Bottom side refers to the side of the section attached to the coverslip, the top side is exposed to buffer. ICC immunocytochemistry. Performed in two biological replicates. Scale bar: **A**: 5 μm, **B**: 2 μm, **C**, **D**: 1 μm.

## Discussion

Microscopy has always represented a fundamental tool for the in-depth analysis of intracellular structures. Unfortunately, the application of advanced imaging techniques in plants (e.g. SMLM or CLEM approaches) lags behind the application in other model systems. This is mainly due to challenges related to the complexity and composition of plant tissues, including the necessity to customize existing protocols or develop entirely new workflows to implement otherwise common labelling methods. Reports of SMLM in plants are rare[26–29], often limited to less demanding high-end techniques such as SIM or 3D-SIM (reviewed in ref. 30). Correlative imaging techniques generally face similar challenges in sample preparation and requirements for protocol optimization. While it has been shown previously that alternate sectioning of embedded plant tissues for fluorescence and electron microscopy can be used for correlative analysis[31], the implementation of super-resolution microscopy in tandem with electron microscopy in plants, especially in conjunction with the unique properties provided

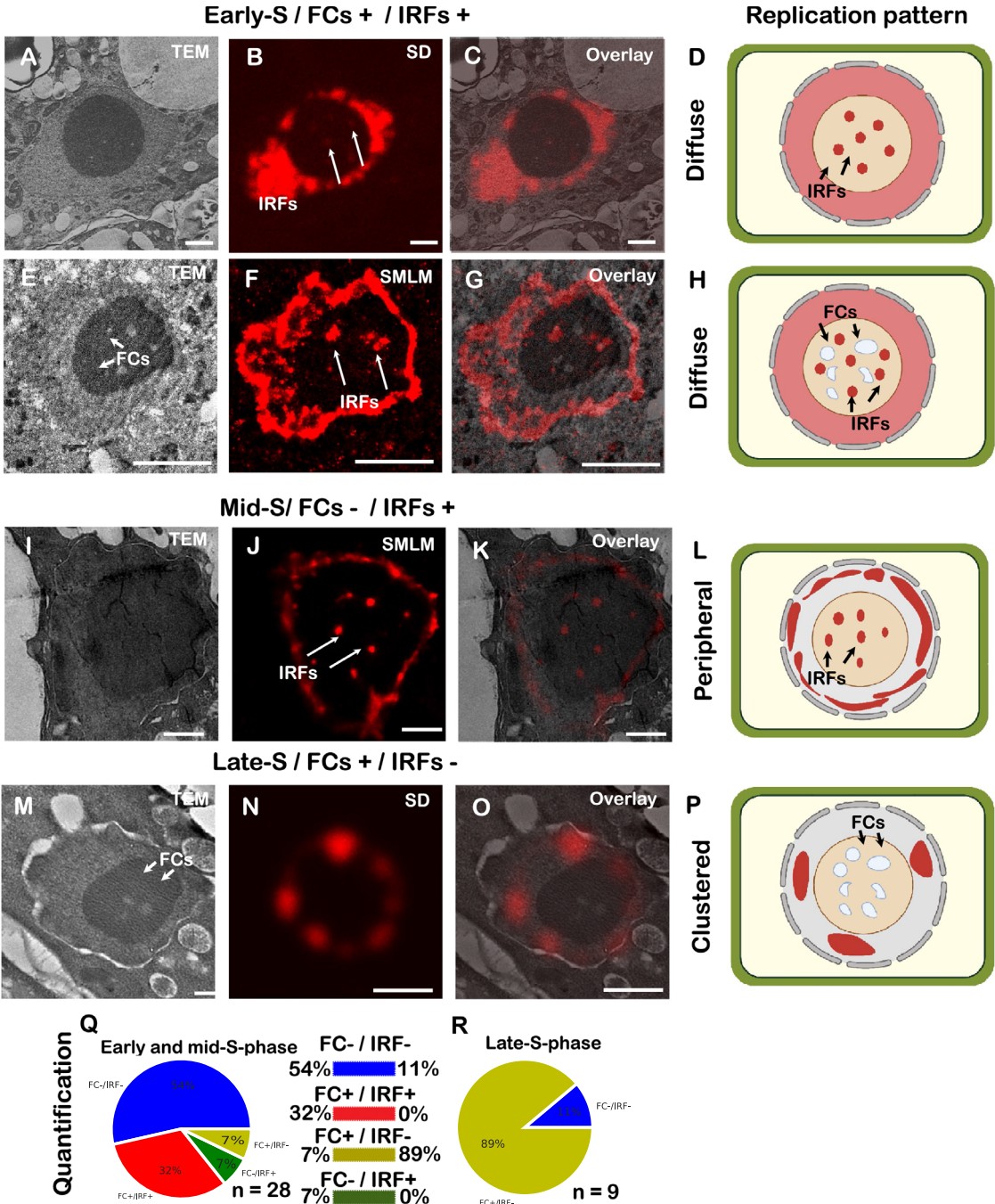

**Fig. 6 | Analysis of the correlation between S-phase progression and nucleolar architecture.** CLEM image of a nucleus in the early-S-phase (**A**–TEM, **B**–SD, **C**–overlay image), with a schematic depiction in (**D**). Examples of early (**E**–**G**) to mid-S (**I**–**K**) phase replicating nuclei (schematically in **H** and **L**, respectively), with and without visible FCs (**E** and **I**, respectively). Examples of late-S (**M**–**O**) phase replicating nuclei (schematically in **P**). Quantification of the replicating profiles (**Q**–early and mid; **R**–late) in relation to the presence of FCs and IRFs, *n* = cells analysed, pooled dataset from both wild-type and mutant (*fas1, nuc1*) plants. SD spinning disk, TEM transmission electron microscopy, FC fibrillar centres, IRFs intranucleolar replication foci, SMLM single-molecule localization microscopy. Scale bar: 2 μm.

by bio-orthogonal chemistry labelling has not been shown previously and is the cornerstone of this study.

Here, we present a workflow for the correlative imaging of plant tissues that combines chemical fixation, sample embedding into an acrylate-based resin (Lowicryl), in-section labelling for super-resolution microscopy and post-contrasting for electron microscopy. The workflow was developed specifically for the analysis of DNA replication and nucleolar organization in plant cells and offers several advantages compared to approaches that use isolated nuclei and conventional labelling methods, both in terms of biological

relevance and technical/imaging capabilities. First of all, imaging of plant sections enables us to localize and identify cells of interest based on the position of the cell within the tissue. In our case, we focused primarily on the nucleolar architecture and replication profiles in cells of the epidermal layers from the meristematic and elongation zones. Second, the sectioning of tissues, coupled with the specific in-section Click-iT labelling results in a lower fluorescence background relative to whole-mount labelling, which is critical for SMLM imaging, as it translates directly into image reconstruction quality and improved resolution.

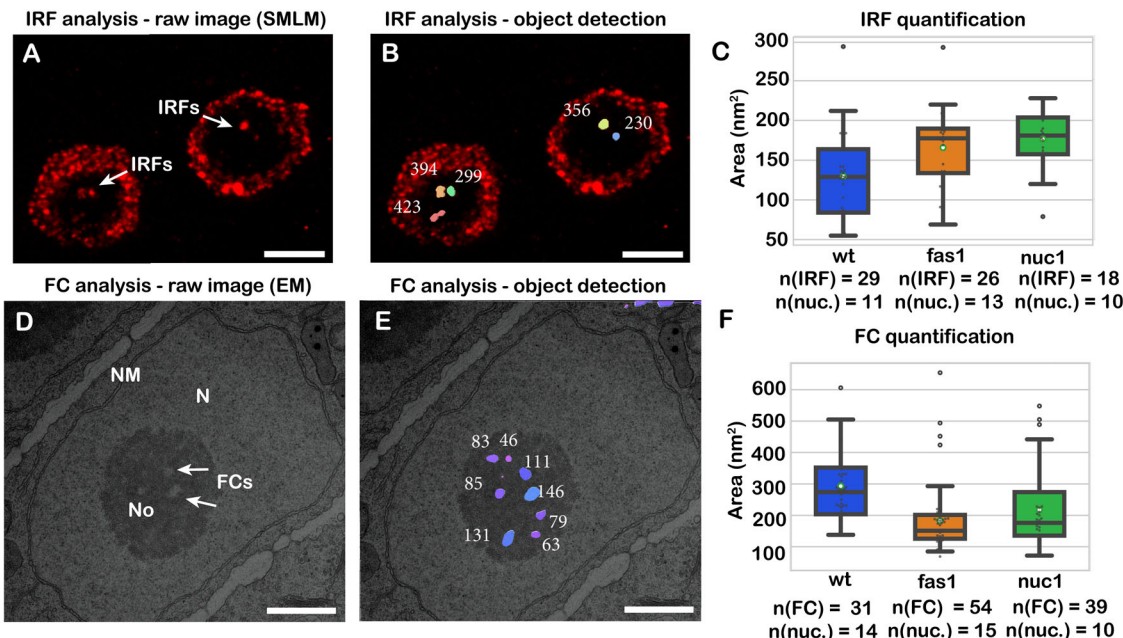

**Fig. 7 | Quantification of the FC and IRF diameter in wild-type and mutant *A. thaliana*.** Quantification of the size of IRFs in SMLM reconstruction (**A**), with structures highlighted after object detection (**B**). Quantification of FCs from TEM micrographs (**D**), with detected structures in (**E**). For IRF and FC quantification (**C**, **F**), *n* = structures analysed. Statistical significance was evaluated based on the non-parametric Kruskal–Wallis H-test (H-statistic = 12.3 for IRF analysis, *p* = 0.0021; H-statistic = 24.1, *p* = 5.83 × 10⁻⁶ for FC analysis; degrees of freedom = 2). Differences between groups were evaluated with the Wilcoxon rank-sum statistic (two-sided). The numbers correspond to the calculated area in nm². Experiments for wild-type, *fas1* and *nuc1* plants were performed in two biological and technical replicates. Experiments were performed in three biological replicates. Means are indicated by white circles, medians are indicated by the line inside the box (with minimum and maximum values are defined by the whiskers, percentiles (25% and 75%) are indicated by the top and bottom edges of the box. Confidence intervals (95%) for the median in the wild-type, *fas1* and *nuc1* IRF and FC measurements respectively are [90:142] for wild-type IRF data, [136:180] for *fas1* IRF, [161:202.5] for *nuc1* IRF, [250:313] for wild-type FC, [135:172.5] for *fas1* FC, [157:229] for *nuc1* FC: **\*** sign indicates statistically significant difference between tested groups (*p* < 0.05). Source data are provided as a Source data file (source data file−sheet 1 for IRF quantification, sheet 2 for FC quantification). FC fibrillar centres, IRFs intranucleolar replication foci. Scale bar (**A**, **B**): 2 μm, (**D**, **E**): 1 μm.

In the context of replication labelling presented in this study, the EdU incorporates selectively into replicating DNA and the subsequent Click-iT reaction mediates the covalent binding of the conjugated fluorophore to ethynyl (for an overview of Click-chemistry, see refs. [32,33]). The incorporation of bio-orthogonal (Click-iT) labelling into the workflow was critical for the SMLM performance so as for the choice of suitable chemical fixation compatible with TEM. Chemical cross-linking of samples with glutaraldehyde is necessary for the preservation of fine subcellular features, however, glutaraldehyde not only introduces some auto-fluorescence but its strong cross-linking properties are often detrimental to immuno-epitopes and reduce both, the efficiency and specificity of immunolabelling by antibodies. Importantly, we found that the Click-iT reaction between the azide and ethynyl moieties is not hindered in samples fixed with up to 3% glutaraldehyde, neither in samples treated with osmium tetroxide (for enhancement of membrane contrast in the TEM) or after sample embedding into the epoxy-based Spurr's resin, although the osmium tetroxide treatment did result in noticeable background interfering with SMLM imaging. Our results suggest that for applications involving bio-orthogonal (Click-iT) labelling and advanced microscopy techniques in this experimental model, the most suitable embedding medium is the acrylate-based polar resin Lowicryl K4M.

In terms of SMLM, Click-iT labelling introduces virtually no linkage error (the distance between the fluorophore tag from the actual position of the epitope), which makes it an optimal strategy for tagging structures. Importantly, Click-iT chemistry is not restricted to labelling nucleic acids and can be successfully applied to tag proteins of interest through the introduction of non-canonical amino acids[34,35]. In recent publications, Click-iT chemistry has been introduced into CLEM workflows and applied to tracking intracellular trafficking or lipids in

bacteria[36,37], showing the promise of this approach in imaging applications. A recent review by Chen et al.[38] summarizes the possible imaging applications in plant sciences, including applications in studying plant cell wall development and structure, showing that the development of novel imaging protocols based on Click-iT chemistry are currently an active area of research.

To demonstrate the utility of the established correlative workflow presented in this work, we examined the nucleolar ultrastructure during replication. Using the CLEM approach in this study was essential to determine whether the presence of the FCs correlates with the detection of IRFs in the early-S-phase. We show that the replication of intranucleolar DNA is restricted to the early phases of S-phase progression, sometimes concomitant with the detection of fibrillar centres. The size of fibrillar centres was decreased in the *fas1* and *nuc1* mutant lines, whereas the number of fibrillar centres per nucleolus was increased in *fas1* plants compared to the wild-type. This suggests that the activity of the nucleolus is maintained, and the smaller size of the FCs is compensated by the increase in the number of FCs, which might be related to the changes in the activation of different rDNA variants in *Arabidopsis* as a consequence of the *fas1* mutation[39]. Whether the decrease in the size of FCs is the cause or the consequence of the increase of the size of intranucleolar replication foci (IRFs) remains to be determined. However, it is possible to conclude that the changes in nucleolar ultrastructure impact physiological processes in the nucleolus, notably replication as detected through DNA replication labelling. We hypothesize that the increase in the size of the IRFs is the result of the relaxed chromatin configuration, related to the altered methylation status of rDNA and its decompaction (*nuc1* mutants; ref. [20]), different levels of histone variants present in rDNA and/or general redistribution of rDNA genes in the nucleolus[15,21]. Our analysis

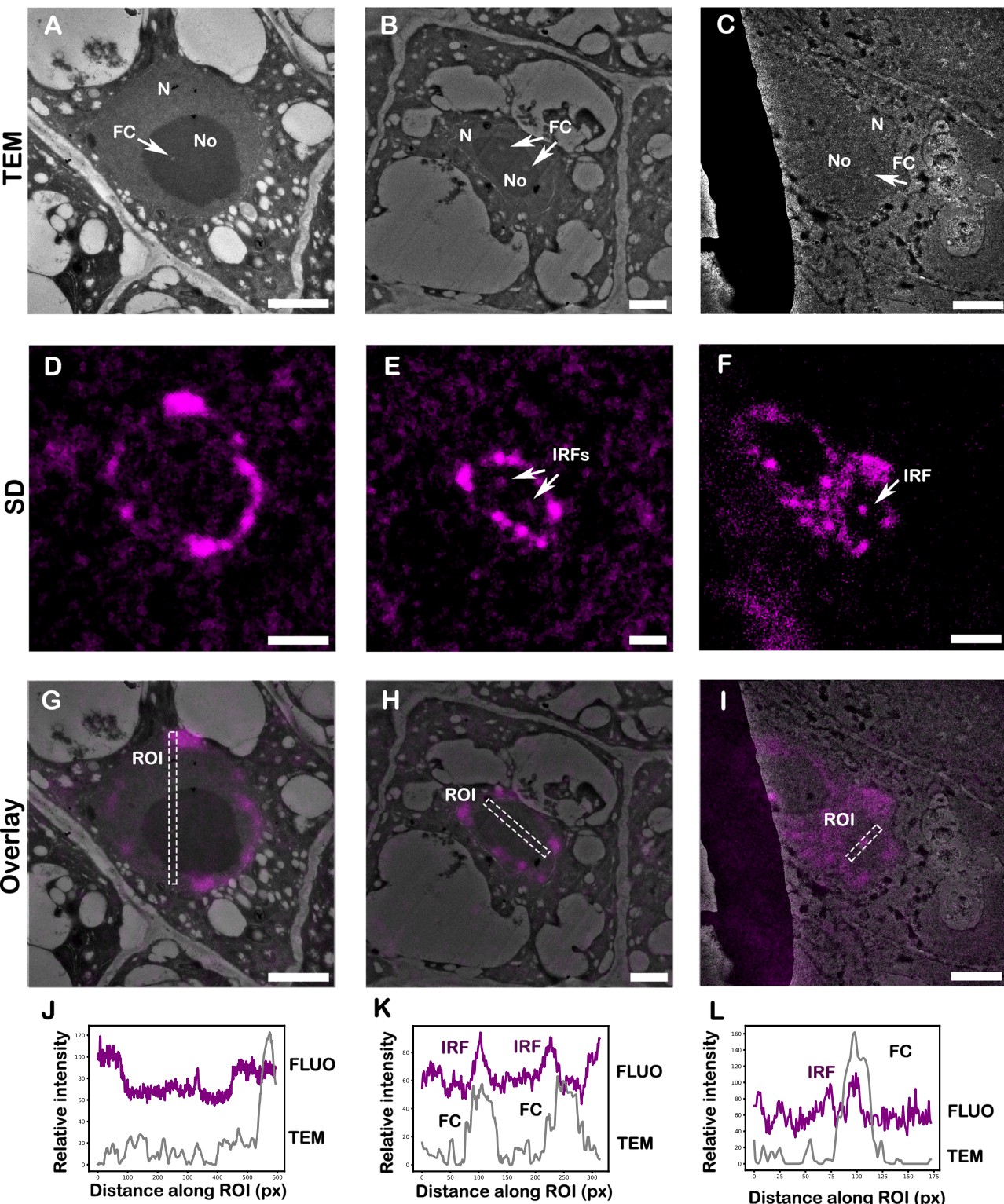

**Fig. 8 | Correlation of the FC and IRF localization on 150 nm sections by CLEM.** Imaging of 150 nm sections on TEM post-contrasted with uranyl acetate and lead citrate on 150 nm sections in the *fas1* mutants (TEM images in **A–C**) with visible nucleus (N), nucleolus (No) and fibrillar centres (FC), marked with a white arrow. Fluorescence imaging of replicating DNA without (**D**) and with intranucleolar replication foci (IRFs−**E**, **F**). The overlay images (**G–I**) and corresponding line profiles (**J–L**) show the level of colocalization between IRFs and FCs. Experiments were performed in two biological and two technical replicates. ROI region of interest, px pixels, TEM transmission electron microscopy, SD spinning disk. Scale bar: 2 μm.

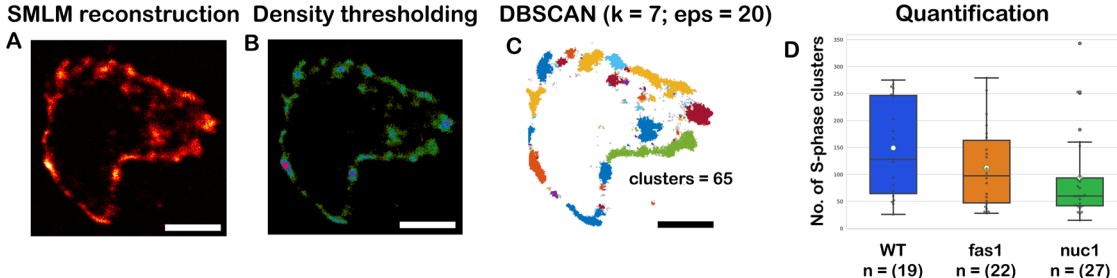

**Fig. 9 | Clustering-based analysis of replication progression on SMLM data.**
Super-resolution images were reconstructed in the SMAP software (**A**). Image thresholding using cluster density calculator (**B**) to eliminate localizations from low-density regions. **C** Detection of clusters using the DBSCAN algorithm. Quantifying the number of clusters in wt, *fas1* and *nuc1* mutants using the Kruskal–Wallis H-test, *n* = number of cells analysed (H-statistic = 5.092; *p* = 0.078; degrees of freedom = 2). **D** Differences between groups have been estimated with the Wilcoxon rank-sum test (two-sided). Experiments were performed in three biological replicates. Means are indicated by white circles, medians are indicated by the line inside the box, minimum and maximum values are defined by the whiskers, percentiles (25% and 75%) are indicated by the top and bottom edges of the box. Confidence intervals (95%) for the median in the wild-type, *fas1* and *nuc1* cluster analysis are [69:245] for wild-type data, [54.5:146] for *fas1* and [46:90] for *nuc1*. Source data are provided as a source data file (sheet 3−DBSCAN clustering analysis). k: minimum objects in the neighbourhood, eps: neighbourhood radius. Scale bar: 1 μm.

of replication foci clustering in the nucleus did not show significant differences between the wild-type and *fas1* or *nuc1* mutant plants, in line with a recent report suggesting that the S-phase progression is not delayed in the *fas1* background[40].

While we believe that the experimental approach presented in this paper is an important advance in correlative and super-resolution imaging of plant nuclei, it has certain limitations. First of all, while we can analyse the same cells in both TEM and SR imaging, there is always an offset in the axial plane during consecutive-section CLEM, since we are imaging two different adjacent sections (500 nm section for SMLM and 70 nm section for TEM). We were able to image 150 nm sections with confocal microscopy for same-section CLEM but did not successfully optimize the imaging of these sections by dSTORM microscopy. Nevertheless, we demonstrated that same-section CLEM is feasible for plant tissue sections embedded in Lowicryl (Fig. 8), essential for correlating the localization of key features. However, there is a trade-off in lower image quality (lower fluorescence intensity on 150 nm sections) and delicate sample handling during repeated washing and drying of sections on EM slot grids. The second important limitation is tied to labelling. We make the case that Lowicryl sections are permeable for low molecular weight fluorophore conjugates, but detecting proteins with classical antibodies is limited since they cannot fully penetrate Lowicryl[41] (Fig. 4F−H). However, this limitation may be overcome by the use of SNAP-tagged or HALO-tagged proteins expressed in plants[42] and their subsequent detection in sections using fluorescently labelled SNAP or HALO ligands characteristic for their low MW and efficient penetration. Besides Click-iT chemistry and SNAP/HALO tag technology, low-molecular weight reagents for labelling other cellular structures (primarily cytoskeleton−SiR and SPY probes; ref. [43]) are available, though the spectrum of applications is limited. Our approach will benefit from the current effort to introduce other labelling strategies which aim to minimize the size of the labelling reagents and thus increase the permeability through Lowicryl resin, including detection with scFv antibody fragments, nanobodies[44,45], proximity labelling with biotin[46] or DAB precipitation-based probes (reviewed in ref. [47]). Lastly, the correlation between SMLM and EM images is not trivial, due to sample deformation and stretching during processing. Partial mismatch of the overlay of the light/fluorescence image (500 nm section) and the corresponding TEM image (70 nm section) in consecutive-section CLEM should also be expected, since both are 2D-projections of a (partial) 3D-volume of a different thickness and the ultrathin TEM section is an adjacent section (continuum) of the semithin section.

Overall, we show that the replication labelling in plant cells is feasible on Lowicryl sections with suitable properties for single-molecule localization microscopy. It allows the analysis of DNA replication in a quantitative manner with the ultrastructure of the nucleolus provided through correlation with TEM imaging, using either consecutive-section or same-section CLEM. We believe the findings in this study will open the path to further applications of resin embedding in tandem with super-resolution microscopy, including relevant plant crop models.

## Methods
### Plant growth and EdU incubation
*Arabidopsis thaliana* seeds (wt Columbia 0, *fas1* SAIL_662_D10 and *nuc1* SALK_053590)[20,48] were sterilized (90% ethanol, 5 min) and plated on half-strength agar Murashige and Skoog medium (½ MS medium) with 1% sucrose. After 1-day stratification (4 °C/dark) plates were transferred to the growth chamber and grown for up to 1 week under long day (LD) conditions (16 h light − 21 °C/8 h dark − 19 °C/50−60% relative humidity). The 7-day-old seedlings of wt Col0, *fas1*, and *nuc1* plants were incubated in 20 μM EdU in 1x liquid MS medium for 90 min and subsequently processed for light and electron microscopy as described below.

### Progressive lowering of temperature and embedding in Lowicryl resin
After the EdU incubation, the samples were rinsed in 1x PHEM buffer (60 mM PIPES, 25 mM HEPES, 10 mM EGTA, 2 mM MgCl₂) pH 6.8 and fixed in 4% formaldehyde (FA, #15710, EMS, Hatfield, USA) and 0.1% glutaraldehyde (GA, #16220, EMS, Hatfield, USA) in PHEM buffer on a rotating platform for 1 h at room temperature, then overnight at 4 °C. After three washes with cold PHEM buffer pH = 6.8 for 10 min, the samples were embedded into 2% low-melting agarose (LMA; 2% agarose in ddH₂O) blocks (according to Wu et al.[23]) and *en block* contrasted with 0.5% uranyl acetate (aqueous solution) overnight at 4 °C. Dehydration and infiltration in Lowicryl resin K4M (#14330, EMS, Hatfield, USA) were performed manually using the progressive lowering of temperature technique in the freeze-substitution unit (Leica EM AFS2). Increasing concentration of ethanol was accompanied by decreasing temperature in the chamber down to −35 °C. During this process, the ethanol was substituted with Lowicryl resin K4M at different ratios (1:1 for 1 h, 1:3 for 1 h), and finally infiltrated three times with 100% Lowicryl K4M. Finally, the samples were polymerized under UV at −35 °C for three days.

### Microwave-assisted processing to Spurr's resin
For the ultrastructural analysis, the samples were rinsed in 1x PHEM buffer and fixed in 3% GA and 0.5% FA in PHEM buffer, pH 6.8. After incubation for 3 hours at room temperature, the samples were fixed in

0.5% GA and 4% FA in PHEM buffer pH 6.8 and stored at 4 °C overnight until further processing. Before embedding to 2% LMA, the samples were washed in cold PHEM buffer at room temperature and all subsequent steps were conducted by a microwave tissue processor (Pelco BioWave Pro+ #3670-230, Redding, USA) equipped with power modulator, vacuum sample container and ColdSpot (system preventing local hotspots generation). All microwave steps (except for dehydration) were performed under vacuum. Samples were first post-fixed with osmium tetroxide (0.5% in milli-Q water) while microwaved at a power of 100 W for 14 min (2 min ON/OFF cycles), then washed in PHEM buffer and contrasted in 0.5% uranyl acetate at a power of 150 W for 7 min. Gradual dehydration in acetone was performed at a power of 250 W for 1 min at each step, followed by infiltration in Spurr's resin (#14300, EMS, Hatfield, USA) first at ratios 1:3, 1:1, 3:1 with acetone, then three times in 100% Spurr's resin (150 W for 7 min each step). The specimens were finally polymerised at 70 °C for 72 h in flat embedding silicon moulds (#70905-01, EMS, Hatfield, USA).

## Collection of semithin and ultrathin sections

For correlative microscopy, cutting and collection of each semithin (500 nm) section was followed by cutting and collection of an adjacent ultrathin (70 nm) section to reduce the Z-offset in image correlation. The ultrathin sectioning was performed in the ultramicrotome (Leica EM UC7) using Ultra Sonic Diamond knife (Diatome, Nidau, Switzerland). The semithin sections were collected onto glass coverslips coated with 0.05% poly-L-lysine (#P8920; Sigma-Aldrich), ultrathin sections on carbon-formvar-coated copper-slot-grids (#G2010-Cu, EMS, Hatfield, USA). Thicker sections (500 nm and 1000 nm) were collected to test resin permeability by labelling for 3D-SMLM. All semithin sections for super-resolution microscopy were labelled using Click-iT chemistry, ultrathin sections and 150 nm sections for TEM were post-contrasted using aqueous 4% uranyl acetate (#22400, EMS, Hatfield, USA) and 3% Reynold´s lead citrate.

## Click-iT labelling, phalloidin staining and immunolabelling

Click-iT detection of EdU on sections attached to poly-L-lysine coated coverslips was performed using the Click-iT cell proliferation kit (#C10340; Thermo Fisher Scientific), according to the manufacturer's protocol. The labelling of 150 nm sections on supporting formvar membranes has been performed by transferring the EM slot on droplets of 1x PBS and the labelling Click-iT solution. For the staining of actin filaments, sections on coverslips were first washed two times in 1x PBST (0.05% Tween-20), blocked for 30 minutes in 3% BSA in 1x PBST and incubated with 150 nM Phalloidin-AF647 (#A22287, Thermo Fisher Scientific) in 3% BSA/PBST for 30 min in a humid chamber. Samples were then rinsed 3x in 1x PBST and imaged in the imaging buffer consisting of 50 mM Tris-HCl, 10 mM NaCl, 50 mM β-mercaptoethylamine (MEA, 30070, Sigma-Aldrich), 1.1 mg/ml glucose oxidase (G2133, Sigma-Aldrich) and 100 µg/ml catalase (C40, Sigma-Aldrich, St. Louis, MO, USA) in Chamlide magnetic holder cells (#CM-B25-1; Live Cell Instrument). For immunolabelling of histone H3, sections on coverslips were first blocked for 30 minutes in 3% BSA in 1x PBST, then incubated with anti-H3 antibody (1:100 dilution, #ab1791, Abcam) in BSA/PBST overnight (#ab1791; Abcam) followed by rinse in 1x PBST and incubation with Alexa Fluor Plus 555 secondary antibody (1:200 dilution, #A32732, Thermo Fisher Scientific) for 45 min at room temperature prior to final wash in 1x PBST.

## Fluorescence microscopy

The images of root sections (500 nm/150 nm) were acquired on the Nikon CSU-W1 confocal spinning disk microscope. Time-lapse dSTORM imaging was performed on a Nikon Ti-E microscope with a Nikon CFI HP Apo TIRF 100x oil objective (1.49 NA, detection with EM CCD Andor iXon Ultra DU897 camera) and on Zeiss Elyra 7 (Carl Zeiss, GmBH) imaging system with a 63x Oil Plan-Apochromat oil objective

(1.46 NA, detection with PCO edge 4.2 sCMOS camera) in the HILO mode. For SMLM of whole-mount samples, roots were immobilized on poly-L-lysine coated high-precision coverslips and mounted in an imaging buffer (as described above). For SMLM of sections, 500 nm and 1000 nm thick root sections were anchored to 22 × 22 mm high-precision poly-L-lysine coated coverslips, mounted in Chamlide holder cells and imaged as described above. For the same-section imaging of 150 nm sections, the EM slot grids were mounted on a coverslip inside the Chamlide holder cell and after fluorescence imaging, the EM slot grid was gently detached from the coverslip after by immersion of the slot in 1x PBS, rinse in distilled water and air-drying for subsequent contrasting and TEM imaging.

## Electron microscopy

TEM images of ultrathin (70 nm) and 150 nm sections were acquired on the Jeol JEM-2100Plus (200 kV) equipped with LaB$_6$ cathode, TVIPS XF416 CMOS 4k x 4k camera and SerialEM software v. 4.0.3[49]. In order to facilitate the localisation of target cells for cross-correlation with images from brightfield and fluorescence microscopy, montages of root tips (each consisting of several hundred images) were collected at pixel size of 2.83 nm, 0.7 s exposure, and 2x binning. Details were recorded with different pixel sizes depending on the region of interest, 2 s exposure and 1x binning. Data were aligned and pre-processed in IMOD software v. 4.11.4[50].

## Image analysis, image alignment and SMLM reconstructions

For SMLM reconstructions, 20,000 time frames were acquired. Reconstructions of super-resolution images were performed using ThunderSTORM[51]. To validate the quality of the SMLM reconstructions by quantitative error mapping of super-resolution images we used the NanoJ-SQUIRREL package[52,53] in FIJI. For Fourier ring correlation (FRC) analysis, 20,000 images from the time series were split into odd and even frames, which were used to reconstruct super-resolution images in ThunderSTORM. Images were then merged into a stack in FIJI and used as the input for the FRC analysis in NanoJ-SQUIRREL. Density-based cluster analysis was performed in the SMAP environment[54]. For the clustering analysis, the super-resolution reconstructions were first processed by the density calculator (counting neighbours for each localization in a circle <12 nm), which was then used to filter out localizations in lowest-density regions. Subsequently, we calculated the number of clusters using DBscan, using the SMAP implementation of the algorithm by Caetano et al.[55]; with the minimum number of objects in the neighbourhood set to 7; the neighbouring radius set to 20. The 3D surface projections were rendered using the Imaris software (Bitplane, Oxford Instruments, v10.0.0). For the analysis of fibrillar centre (FC) and intranucleolar replication foci (IRF) size, we used automatic object detection in the Imaris software, where we first performed image smoothing (surface detail = 0.05 µm) followed by thresholding based on background subtraction. The identified intranucleolar objects were then manually selected on each image and their area was used for statistical analysis, as described below. For the image alignment of confocal and EM data, we used the ec_CLEM plugin in Icy[24], with the manual input of fiducial landmarks, using the affine transformation and anisotropic noise model. For dSTORM reconstructions where the transmitted light data was not acquired, we overlaid the images manually using the Zen Connect suite, with linear transformations such as isometric image stretching and rotation.

## Statistics and reproducibility

For quantitative analysis, we pooled data from two biological replicates (e.g. two different seedlings/condition) and two technical replicates for each biological replicate (e.g. different sections contrasted/labelled on a coverslip or EM grid), with the following exceptions. Figure 3 displays the general workflow, which has been

replicated five times. 3D dSTORM microscopy in Fig. 4 was repeated in two technical replicates. Results in Fig. 5 have been replicated in two biological replicates. Results in Fig. 6 are from the pooled dataset of wild-type, *fas1* and *nuc1* plants (all of which have been imaged in 2 biological replicates). Results in Fig. 9 have been obtained from three biological replicates. For the statistical analysis of the size of intra-nucleolar replication foci (IRF) and fibrillar centres (FCs), we applied the non-parametric Kruskal–Wallis H-test and subsequently the two-sided Wilcoxon rank-sum post hoc test. H-values, *p*-values and adjusted *p*-values for post hoc tests are presented in the text. Data normality was checked using the Shapiro–Wilk test for normality. Estimation of confidence intervals was performed using a boot-strapping method (Scipy.Stats.Bootstrap). Statistical analyses were performed in the SciPy python library. No statistical method was used to predetermine sample size. No data were excluded from the analyses. The experiments were not randomized, as no experimental treatment was administered to the samples. The investigators were not blinded to allocation during experiments and outcome assessment.

### Reporting summary
Further information on research design is available in the Nature Portfolio Reporting Summary linked to this article.

### Data availability
The raw datasets of TEM imaging (Spurr and Lowicryl) and SMLM data for quantitative analysis have been deposited to the BioImage Archive, under accession code S-BIAD700 (https://www.ebi.ac.uk/biostudies/bioimages/studies/S-BIAD700). Source data are provided with this paper (sheet 1—IRF quantification, sheet 2—FC quantification, sheet 3—DBSCAN analysis). Source data are provided with this paper.

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

## Acknowledgements

We acknowledge the core facilities CELLIM and BioCEV supported by the Czech-BioImaging large RI project (LM2023050 funded by MEYS CR - L.K., J.M., D.L., J.P., M.E. and E.M.) for its support in obtaining scientific data presented in this paper as well as the European Regional Development Fund projects No. CZ.02.1.01/0.0/0.0/16_013/0001775, CZ.02.1.01/0.0/0.0/18_046/0016045 and CZ.02.01.01/00/23_015/0008205 (M.E.). The project was funded by the Czech Science Foundation (project No. 23-06643S to M.D., and GACR-EXPRO project 20-01331X to M.F. and J.F.). We acknowledge the Czech Bioimaging initiative for providing access and financial support for advanced imaging techniques. Figures 1 and 6 have been co-created with BioRender.com.

## Author contributions

M.F., L.K., J.M. and D.L. performed all experiments and sample preparation. M.F., L.K., J.M., J.P. and E.M. performed electron and super-resolution microscopy. M.F. conducted data analysis. M.F., L.K., D.L., M.E., M.D. and J.F. performed experimental design, manuscript preparation and provided technical background and resources for the study.

## Competing interests

The authors declare no competing interests.
