## [Peer Review File · Nature Communications]

REVIEWER COMMENTS

Reviewer #1 (Remarks to the Author)

Correlative light and electron microscopy is a useful tool that combines the unique benefits of light and electron microscopy. However, it also entails challenges to be widely used in plant biology because of the permeability of plant tissues, fluorescence labelling efficiency, and indexing of features of interest. In this manuscript, the authors established a correlative workflow for *Arabidopsis* roots, which enables Click-iT labelling on Lowicryl sections and is compatible with single-molecule localization microscopy. They then used this system to observe the DNA replication sites and analyze the S-phase progression and nucleolar organization in wildtype and two *Arabidopsis* mutants. These results advance the visualization of ultrastructural features of plant tissues. As mentioned by the authors, however, there are several limitations of the correlative workflow and needs to be improved. Furthermore, they didn't provide too much insight into unknown phenomenon or mechanism.

Labeling efficiency is an important parameter in evaluation of the feasibility of the correlative workflow, and antibodies are widely used to detect proteins of interest. However, detecting proteins with classical antibodies is limited in this study. Although the authors discussed potential method such as using SNAP-tagged or HALO-tagged proteins expressed in plants to tackle this limitation, I would suggest the authors try this method to demonstrate the feasibility of low molecular compound labelling in this workflow, which would benefit the application of this workflow by other researchers.

In Fig. 3C, one of four cells could overlay in spinning-disk fluorescence and transmission electron microscopy. The authors should mention details of the percentage of cells that could overlay in both fields.

Fig. 5 The structure of the nucleic in B1-B3 looks like mid-S phase nuclei. Please check and clarify how the authors differentiated the early-S and mid-S phase nuclei. Line 345, the authors mentioned that dataset are pooled from both wildtype and mutant plants. It is confusing which data is obtained from wildtype plant or mutant plants. Line 335, Fig5D is not referred correctly. Fig 5D, 5E, and 5F are not referred in the text.

Fig. 6 The size of IRFs and FCs in a single nucleus varied. It is confusing how the size of IRFs and FCs are calculated. How many nuclei are analyzed? Is the number of IRFs and FCs different in wildtype and mutant plants?

Line 258-261, move this sentence to next paragraph.

Line 314-315, move this sentence to next paragraph.

Fig.4 A-D and E to H are described in different sections of the Results. I suggest the authors split the figures to two independent figures.

Reviewer #2 (Remarks to the Author)

Franek et al. use super-resolution CLEM to image plant biology with both SMLM and TEM. Fig 1 provides a clear schematic of the workflow, and this appears to be an impressive approach to an impressive application. However, the authors are actually doing serial section imaging, not “true” CLEM, where the same sample is imaged by both modalities. This reduces the impact significantly, as serial-section CLEM has been possible for decades and the only addition here is super-resolution (which has also been done on sections before). This is not to say the work isn’t impressive – it is! – but that there is unfortunately reduced novelty. (However, I am not an expert in plant biology, so do not comment on that aspect of that work or the application thereof.)

Consequently, I do not believe this work fits Nature Communications, but would be suitable for publishing as is in, e.g., Scientific Reports.

I have one major request:

Information on image alignment/correlation is missing. What software was used? Were transforms limited to shift/rotation? Affine? Free-form? These are important parameters as they greatly affect correlation accuracy, especially when dealing with SMLM reconstructions.

Other points:

The authors might like to consider including the following apposite references:

Osuga et al. Mol Biol Cell. 2021 Nov 1; 32(21): br7.

Development of a green reversibly photoswitchable variant of Eos fluorescent protein with fixation resistance

Paez-Segala et al. Nature Methods volume 12, pages215–218 (2015)

Fixation-resistant photoactivatable fluorescent proteins for CLEM

Line 18: “is an essential tool” – this is too strong a statement in this context.

Figure legend 3: “Insets, highlighted by red rectangles” add text to indicate rectangle is in panels A and B.

Reviewer #3 (Remarks to the Author)

“In-section super-resolution CLEM: Shedding light on nucleolar structure and replication in plants” by Dr. Franek and colleagues introduces a correlative workflow for the imaging of plant cells with single molecule localization microscopy (SMLM) and electron microscopy (EM). The authors use this workflow to characterize DNA replication in *Arabidopsis thaliana*.

Key results of the study include:

-The possibility to perform on-section-labelling for SMLM using Click-iT chemistry after embedding in EM resin.

-The feasibility of performing on section CLEM with SMLM on lowicryl embedded sections, with the possibility of 2D and 3D SMLM.

- The co-occurrence of fibrillar centers (FCs, deduced from TEM) and intranucleolar replication foci (IRFs, deduced from SMLM) in early-S-phase cells.

-Changes of the nucleolar ultrastructure of mutant *Arabidopsis thaliana*.

The article is well written and constitutes an important contribution to the field of correlative microscopy. The authors not only present a workflow for advanced CLEM but also demonstrate how it can be used in an application that relies on correlative TEM and SMLM imaging. The statements made about DNA replication in *Arabidopsis thaliana* are supported by a valid statistical analysis. I recommend the article for publication with two general remarks and a few minor comments that should be addressed.

General remarks:

Generally, the figures are well arranged and every key point is supported by corresponding micrographs. However, the size of some figures is a bit small, especially the EM images would profit from a larger format.

Ultrastructural preservation is the key to any EM study. Please comment on the structural preservation of your samples. Higher magnification TEM images should be provided either in the article or in the supplementary material to assess the preservation of cellular ultrastructure using the described approaches.

Minor comments:

l.114/115: "During this process, the ethanol was substituted with LowicrylK4M resin at different ratios, (...)" Please specify which ratios of lowicryl were used.

l. 137: "Diatome, Hatfield, Switzerland" should be changed to either "Diatome, Hatfield, USA" for the US branch of the company, or "Diatome, Nidau, Switzerland" to refer to the European branch.

Figure 2: Please consider adding scalebars to the overlay images in e1, e2, and e3.

Figure 3: Please consider adding scalebars to C, D, and E.

l. 337: Please specify what you mean with "partial colocalization". As for now, I can not see any further colocalization between the IRFs in the SMLM image and the FCs in the TEM image than their occurrence in the same cell, which is already mentioned in the first part of the sentence. If a statement about colocalization is made, please discuss the influence of sample stretching and distortion as described earlier in the article (e.g. figure 3), as well as z-axial offset.

Figure 5: A spinning disk (SD) micrograph is shown as an example for a cell not containing an intranucleolar replication focus (IRF) in figure 5 A2 and A3. Please either use a SMLM image to show such a cell (as in B2 and C2), or show an SD image of a cell containing an IRF for comparison.

Figure S3: The numbers of inset 1 and 2 are switched in comparison to the overview. Please change the figure accordingly.

Response to reviewers:

Reviewer 1:

Correlative light and electron microscopy is a useful tool that combines the unique benefits of light and electron microscopy. However, it also entails challenges to be widely used in plant biology because of the permeability of plant tissues, fluorescence labelling efficiency, and indexing of features of interest. In this manuscript, the authors established a correlative workflow for Arabidopsis roots, which enables Click-iT labelling on Lowicryl sections and is compatible with single-molecule localization microscopy. They then used this system to observe the DNA replication sites and analyze the S-phase progression and nucleolar organization in wildtype and two Arabidopsis mutants. These results advance the visualization of ultrastructural features of plant tissues. As mentioned by the authors, however, there are several limitations of the correlative workflow and needs to be improved. Furthermore, they didn't provide too much insight into unknown phenomenon or mechanism.

Q: Labeling efficiency is an important parameter in evaluation of the feasibility of the correlative workflow, and antibodies are widely used to detect proteins of interest. However, detecting proteins with classical antibodies is limited in this study. Although the authors discussed potential method such as using SNAP-tagged or HALO-tagged proteins expressed in plants to tackle this limitation, I would suggest the authors try this method to demonstrate the feasibility of low molecular compound labelling in this workflow, which would benefit the application of this workflow by other researchers.

A: While we agree that SNAP / HALO labelling would be beneficial for the study and would further demonstrate the utility of the approach, generating SNAP / HALO tagged stably expressing plants is not feasible within the timeframe of the review (the process itself takes from 6 - 12 months). For this reason, we tested phalloidin labelling instead, which is another low molecular weight compound that demonstrates the crucial point that the Lowicryl resin is permeable for low molecular weight compounds such as Alexa Fluor 647 azide and Phalloidin - Alexa Fluor 647.

In Fig. 3C, one of four cells could overlay in spinning-disk fluorescence and transmission electron microscopy. The authors should mention details of the percentage of cells that could overlay in both fields.

A: We first calculated the percentage of cells tagged with replication labelling on 500 nm sections, to see what percentage of cells are labelled in a small volume in the Z-dimension. We found that 26.1% percent of cells (n = 330 cells from 4 distinct sections) visible in the field of view are in the S-phase (labelled with click-iT chemistry). The ratio of tagged cells that have both visible nuclei in the EM micrographs (from the overview EM images) and visible replication labelling (from the corresponding overview confocal micrographs of the same adjacent section) was 65.2%, which suggests approximately a third of the cells cannot be used for LM-EM correlation in consecutive-section CLEM, due to the offset in the Z-axis. This information was added to the manuscript (page 6 - highlighted in green).

Fig. 6 The structure of the nucleic in B1-B3 looks like mid-S phase nuclei. Please check and clarify how the authors differentiated the early-S and mid-S phase nuclei. Line 345, the authors mentioned that dataset are pooled from both wildtype and mutant plants. It is confusing which data is obtained from wildtype plant or mutant plants. Line 335, Fig6D is not referred correctly. Fig 6D, 6E, and 6F are not referred in the text.

A: As the reviewer points out correctly, the nucleus on the image B1-B3 looks more like a mid-S phase nucleus, this was amended in the text. We have previously published articles (Dvorackova et al., 2018) examining the discrimination of early / mid / late-S phase replicating nuclei, though differentiating early from mid-S phase can be difficult, which we mention in the Discussion section of the manuscript. We now added an explicit statement about the criteria for discriminating early / mid / late S phase nuclei in Results - page 9, highlighted in green, though for the analysis of the presence of IRFs correlating the presence of the FCs, we pooled the early and mid S-phase nuclei. We also added a schematic representation of the replication profiles in Fig. 6 A4, B4, C4, D4 for clarity. We further argue that the replication profiles obtained from SMLM reconstructions look different from confocal / widefield images and there are not many publications on the topic in plants, which makes distinguishing early S from mid S more difficult.

Q: “the authors mentioned that dataset are pooled from both wildtype and mutant plants. It is confusing which data is obtained from wildtype and mutant plants”

A: As mentioned in the text, the consecutive-section analysis of the presence of IRFs / S-phase stage and FC presence is from the entire pooled dataset - meaning wild-type plants, *fas1* plants and *nuc1* plants altogether. This is due to the fact that a) separate EM or LM datasets do not indicate that any of the mutants has absent FCs or IRFs and b) the acquisition of a sufficient size dataset for each individual mutant is outside of the scope of the article, given the fact that the article is primarily a technical / methodological resource.

Q: Line 335, Fig6D is not referred correctly. Fig 6D, 6E, and 6F are not referred in the text.

Figure 6 has been changed, with the addition of an illustration for different replication patterns (which also addresses the previous reviewer's comment about discrimination criteria for the stages of the S-phase). The figure is now referred to correctly in the text.

Q: Fig. 7 The size of IRFs and FCs in a single nucleus varied. It is confusing how the size of IRFs and FCs are calculated. How many nuclei are analyzed? Is the number of IRFs and FCs different in wildtype and mutant plants?

A: In the original version of the manuscript, we measured the size of the IRFs and FCs by measuring the diameter of the objects using the line profile tool in ImageJ. For the revised version, we re-analyzed the data from scratch with automatic detection in the Imaris suite. We first applied smoothing to the images (surface detail = 0.05 μm), then thresholded the images based on background subtraction (for both the FCs and IRFs) in the Imaris 10.0 image analysis software and then calculated the area of intranucleolar objects (FCs and IRFs, Figure 7B, 7E), detected by the surface detector plugin. The threshold was set the same for all the images, so the bias in the evaluation should be minimal in this case. The details of the analysis have been included in the materials and methods section (page 4 and 5, highlighted in green). We also updated the statistics to include the number of nuclei and the average number of FCs /IRFs per nucleolus, this information was included in the manuscript with a significant finding that the number of FCs per nucleolus differs between wt and *fas1* plants, which is further addressed in the discussion.

Q: Line 258-261, move this sentence to next paragraph.

A: Moved to the next paragraph as requested

Q: Line 314-315, move this sentence to next paragraph.

A: Moved to the next paragraph as requested

Fig.4 A-D and E to H are described in different sections of the Results. I suggest the authors split the figures to two independent figures.

A: The figure has been split into two different figures (Fig. 4 and Fig. 5) as suggested.

Reviewer 2:

Franek et al. use super-resolution CLEM to image plant biology with both SMLM and TEM. Fig 1 provides a clear schematic of the workflow, and this appears to be an impressive approach to an impressive application. However, the authors are actually doing serial section imaging, not “true” CLEM, where the same sample is imaged by both modalities. This reduces the impact significantly, as serial-section CLEM has been possible for decades and the only addition here is super-resolution (which has also been done on sections before). This is not to say the work isn’t impressive – it is! – but that there is unfortunately reduced novelty. (However, I am not an expert in plant biology, so do not comment on that aspect of that work or the application thereof.) Consequently, I do not believe this work fits Nature Communications, but would be suitable for publishing as is in, e.g., Scientific Reports.

A: In terms of correlative microscopy nomenclature, we note that even in one of the publications suggested by Reviewer 2, the authors mention “consecutive section CLEM” (Paez-Segala et al., 2015, Nature Methods), which suggests that while different sections are used for EM and LM imaging, the approach is still considered a correlative microscopy “CLEM” approach. As such, we decided to use this nomenclature in the relevant parts of the manuscript, (changes highlighted in green). Prompted by the very relevant comment concerning the same-section CLEM approach, we successfully tested and implemented same-section CLEM on 150 nm sections in our workflow, and integrated these results into the manuscript. As stated in the discussion, same-section CLEM implementation is relevant for the mapping of IRF and FC colocalization, however the consecutive-section CLEM yields better results in quantitative analysis of IRF size as well as the ultrastructural mapping of chromatin, since we did not successfully optimize dSTORM microscopy on 150 nm sections on EM slots.

In terms of novelty, we assume that imaging the same sample (section) by both modalities *per se* would not be a novelty either (as there is plenty of published work on such a workflow). Yet what we do consider as a novelty here is the possibility of using “in-section labelling” using low-MW compounds for correlative imaging, because this can solve significant technical constraints (especially in plant samples) one has to face during more conventional in-resin fluorescence labelling (labelling before sample processing and embedding in resin).

I have one major request:

Information on image alignment/correlation is missing. What software was used? Were transforms limited to shift/rotation? Affine? Free-form? These are important parameters as they greatly affect correlation accuracy, especially when dealing with SMLM reconstructions.

A: We used both the manual alignment of images in Zen Connect and image alignment using `ec_CLEM` open-source software integrated into Icy (Paul-Gilloteaux et al., 2017) in cases where we aligned dSTORM reconstructions and confocal data with EM images, respectively. For the alignment of images, we used the affine transformation model and the anisotropic noise model, with the manual input of fiducial points as landmarks targeted on the edges of cell walls visible in transmitted light and EM micrographs. An example of the semi-automatic alignment with `ec_CLEM` has been introduced in Supplementary Figure 1. The image alignment description has been added to materials and methods, page 4, highlighted in green. At the outset, we considered using fiducial markers, but this option does not seem optimal for tissue sections, as the fiducial markers would be only on the periphery of the roots, which would not help image registrations inside the tissue. As the reviewer points out, another issue with the correlation with fiducials would be the size. Small fiducials might penetrate the tissue but would be lost in correlation due to the axial offset, while large fiducials would not penetrate the tissue.

The authors might like to consider including the following apposite references:

Osuga et al. Mol Biol Cell. 2021 Nov 1; 32(21): br7.

Development of a green reversibly photoswitchable variant of Eos fluorescent protein with fixation resistance

Paez-Segala et al. Nature Methods volume 12, pages215–218 (2015)
Fixation-resistant photoactivatable fluorescent proteins for CLEM

A: We have included the references in the introduction (page 2, highlighted in green).

Q: Line 18: “is an essential tool” – this is too strong a statement in this context.

Figure legend 3: “Insets, highlighted by red rectangles” add text to indicate rectangle is in panels A and B.

A: Line 18: changed to “ is an important tool”; Figure legend 3 changed to: “Insets, highlighted by red rectangles in the panels A and B”.

Reviewer 3:

“In-section super-resolution CLEM: Shedding light on nucleolar structure and replication in plants” by Dr. Franek and colleagues introduces a correlative workflow for the imaging of plant cells with single molecule localization microscopy (SMLM) and electron microscopy (EM). The authors use this workflow to characterize DNA replication in *Arabidopsis thaliana*.

Key results of the study include:

-The possibility to perform on-section-labelling for SMLM using Click-iT chemistry after embedding in EM resin.

-The feasibility of performing on section CLEM with SMLM on lowicryl embedded sections, with the possibility of 2D and 3D SMLM.

- The co-occurrence of fibrillar centers (FCs, deduced from TEM) and intranucleolar replication foci (IRFs, deduced from SMLM) in early-S-phase cells.

-Changes of the nucleolar ultrastructure of mutant *Arabidopsis thaliana*.

The article is well-written and constitutes an important contribution to the field of correlative microscopy. The authors not only present a workflow for advanced CLEM but also demonstrate how it can be used in an application that relies on correlative TEM and SMLM imaging. The statements made about DNA replication in *Arabidopsis thaliana* are supported by a valid statistical analysis. I recommend the article for publication with two general remarks and a few minor comments that should be addressed.

General remarks:

Q: Generally, the figures are well arranged and every key point is supported by corresponding micrographs. However, the size of some figures is a bit small, especially the EM images would profit from a larger format.

A: We enlarged the size of the EM images (e.g. Figure 3 - D, E, F, H, I; Figure 6 - A, B, C, D; Figure 7 - D, E, Supplementary figure S4), and also re-exported all the EM images in a different format, as we noticed the importing of TIFF files to photoshop leads to lower image quality. In supplementary figure 4, we changed the images demonstrating WT and *nuc1* architecture for different images from same dataset captured at a higher magnification, for better clarity.

Q: Ultrastructural preservation is the key to any EM study. Please comment on the structural preservation of your samples. Higher magnification TEM images should be provided either in the

article or in the supplementary material to assess the preservation of cellular ultrastructure using the described approaches.

A: We have added a supplementary PDF file (Supplementary file 1, page 13 in the manuscript) which is a gallery of high magnification EM images of nuclei from Wt, *fas1* and *nuc1* samples in either Lowicryl or Spurr embedding, with marking of cellular structures.

Minor comments:

I.114/115: "During this process, the ethanol was substituted with LowicrylK4M resin at different ratios, (...)" Please specify which ratios of lowicryl were used.

A: This information has been added to the manuscript in the Materials and Methods section.

I. 137: "Diatome, Hatfield, Switzerland" should be changed to either "Diatome, Hatfield, USA" for the US branch of the company, or "Diatome, Nidau, Switzerland" to refer to the European branch.

A: Changed as requested.

Figure 2: Please consider adding scalebars to the overlay images in e1, e2, and e3.

A: Scale bars have been added to the figure.

Figure 3: Please consider adding scalebars to C, D, and E.

A: Scale bars have been added to the figure.

I. 337: Please specify what you mean with "partial colocalization". As for now, I can not see any further colocalization between the IRFs in the SMLM image and the FCs in the TEM image than their occurrence in the same cell, which is already mentioned in the first part of the sentence. If a statement about colocalization is made, please discuss the influence of sample stretching and distortion as described earlier in the article (e.g. figure 3), as well as z-axial offset.

A: Based on new experiments conducted on 150 nm sections (same-section CLEM), we discuss this issue in the corresponding passages in the manuscript (tied to Fig. 8).

Figure 5: A spinning disk (SD) micrograph is shown as an example for a cell not containing an intranucleolar replication focus (IRF) in figure 6 A2 and A3. Please either use a SMLM image to show such a cell (as in B2 and C2), or show an SD image of a cell containing an IRF for comparison.

A: We changed Figure 6 to include a CLEM image showing IRFs seen in the spinning-disk modality (we refrained from imaging late-S-phase nuclei with super-resolution microscopy as the added value in imaging dense chromatin clusters is minimal).

Figure S3: The numbers of inset 1 and 2 are switched in comparison to the overview. Please change the figure accordingly.

A: The error has been corrected and the scale bars have been added to the inset image.

REVIEWER COMMENTS

Reviewer #1 (Remarks to the Author):

Although the authors have addressed most of the issues, I still think that the reviewers do not provide too much insight into unknown mechanism, which reduces the novelty of this manuscript. The other major issue is that potential application of the improved method. Although the authors also used phalloidin to label actin, only limited labelling reagents are available for most proteins or cellular structures. This reduces the significance when compared to the established literature.

Reviewer #2 (Remarks to the Author):

The authors have adequately addressed my concerns.

Reviewer #3 (Remarks to the Author):

I am convinced that the revisions presented by Dr. Franek and colleagues pose a big improvement on the already high quality of the manuscript. Especially the demonstration that their workflow also allows imaging with same-section CLEM is an important addition, since it eliminates the concerns regarding z-axial offset in consecutive section CLEM.

Most of my comments have been dealt with in a satisfactory manner, I only have one concern left:

In response to my previous comments, the authors added a supplementary file. This file is a welcome addition that demonstrates the ultrastructural preservation of the samples by showing TEM images of lowicryl and Spurr embedded samples.

As for now, the only reference to this supplementary file is in a section that deals with changes in the nucleolar ultrastructure in *Arabidopsis thaliana* mutants. While it makes sense to reference the file here, I was a bit confused that this is the only mention of the file.

In a methodological context, I would expect a reference to the supplementary file (alongside a more detailed discussion of the – in my opinion quite decent – ultrastructural preservation) in the first part of the results section, together with the general evaluation of the CLEM workflow. Knowing the qualities and limitations of the workflow presented here will help other researchers to decide whether it is feasible to adapt it for their own studies that might focus on different cellular structures. Therefore, it might be beneficial to include higher magnification TEM images of lowicryl and Spurr embedded samples showing close-ups of different organelles, e.g. a mitochondrion or endoplasmic reticulum. Also, scale bars should be added to all images of the supplementary file.

Response to reviewers

Reviewer #1 (Remarks to the Author):

Although the authors have addressed most of the issues, I still think that the reviewers do not provide too much insight into unknown mechanism, which reduces the novelty of this manuscript.

A: We accept the comment that the manuscript doesn't necessarily provides insight into an unknown mechanism, this is not something that is argued in the manuscript nor was primary focus. To uncover unknown mechanisms for physiological processes that we study (e.g. nucleolar architecture and replication), the experimental design would have to be different from the onset – these studies usually require multi-modal approaches (e.g. microscopy + biochemistry + functional studies etc.). The paper is submitted as a Methodology paper in microscopy, it describes a new protocol and its application in a particular setup. Our approach improves super-resolution fluorescence imaging of plant tissues and enables us to perform correlative microscopy without sacrificing ultrastructural preservation (as we now highlight in the added supplementary figure 1) or the need to optimize the protocol to preserve the fluorescence in resin to perform in-resin CLEM. In this case, it is the advance in correlative microscopy in plants, namely the applicaiton of super-resolution microscopy in Lowicryl sections, possibility of consecutive-section and same-section CLEM and its usefulness in studying the nucleolar archicture in parellel with replication progression.

The other major issue is that potential application of the improved method.

A: We argue that the finding that Lowicryl embedding is compatible with Click-iT chemistry and low molecular weight compound labelling could have broad applications. The Click-iT chemistry tagging of proteins has been already developed by the introduction of non-canonical aminoacids (Nikic et al., 2015), and HALO-tag / SNAP-tag chemistry works on the same underlying principle – e.g. a larger label on the molecule of interest introduced prior to the fixation and embedding and subsequent detection with a low molecular weight compound. The bottleneck in both cases is in the time preparing the constructs and their introduction to a system (e.g. cloning of a HALO-tag protein and stable transformation into cells). We also envision that tags for Click-iT chemistry can be introduced to probes for the detection of specific genomic segments during fluorescence in-situ hybridization (FISH), though this would require non-denaturing FISH approaches. The general idea of in-resin labelling through Click-iT chemistry or HALO/SNAP-tag chemistry is the key point and is not restricted to a particular model – this technique could be used for tissues sections from any model, or from whole sample embeddings of small model organisms (e.g. *Caenorhabditis elegans* / *Drosophila melanogaster*). For plant science, the applications of Click-iT chemistry have been recently reviewed, showing promising new applications in studying plant growth, metabolism, or immune responses (Chen et al., 2023). This reference has been included in the manuscript.

Although the authors also used phalloidin to label actin, only limited labelling reagents are available for most proteins or cellular structures. This reduces the significance when compared to the established literature.

A: We agree that non-specific labelling of cellular structures is limited to an extent. However, there are some options for non-antibody based labelling of structures. Membranes and exosomes can be tagged by lipophilic compounds such as PHK67 and cytoskeleton by compounds such as SiR-Tubulin (Li et al., 2022) or SPY-Actin (which are low-molecular weight compounds - fluorescently labelled derivatives of drugs binding actin / tubulin). As mentioned above, it should be possible to tag any protein of interest by the application of HALO/SNAP-tag technology, with high permeability in Lowicryl for the detection reagents. Importantly, the labelling with antibodies is also possible as we show in the manuscript for the histone H3, even though the permeability is lower. We also note that novel labelling strategies are in development that employ antibody fragments (e.g. scFv antibodies – Han et al., 2023 on Biorxiv) or nanobodies (de Beer and Giepmans, 2020; Fang et al., 2018) for CLEM, which should have higher permeability into the Lowicryl resin. We have added the possible application of SiR-Tubulin/SPY-Actin as well as nanobody techniques in the discussion. Line number 624 – highlighted in green.

Reviewer #2 (Remarks to the Author):

The authors have adequately addressed my concerns.

Reviewer #3 (Remarks to the Author):

I am convinced that the revisions presented by Dr. Franek and colleagues pose a big improvement on the already high quality of the manuscript. Especially the demonstration that their workflow also allows imaging with same-section CLEM is an important addition, since it eliminates the concerns regarding z-axial offset in consecutive section CLEM.

Most of my comments have been dealt with in a satisfactory manner, I only have one concern left:

In response to my previous comments, the authors added a supplementary file. This file is a welcome addition that demonstrates the ultrastructural preservation of the samples by showing TEM images of lowicryl and Spurr embedded samples.

As for now, the only reference to this supplementary file is in a section that deals with changes in the nucleolar ultrastructure in *Arabidopsis thaliana* mutants. While it makes sense to reference the file here, I was a bit confused that this is the only mention of the file.

In a methodological context, I would expect a reference to the supplementary file (alongside a more detailed discussion of the – in my opinion quite decent – ultrastructural preservation) in the first part of the results section, together with the general evaluation of the CLEM workflow. Knowing the qualities and limitations of the workflow presented here will help other researchers to decide whether it is feasible to adapt it for their own studies that might focus on different cellular structures. Therefore, it might be beneficial to include higher magnification TEM images of lowicryl and Spurr embedded samples showing close-ups of different organelles, e.g. a mitochondrion or endoplasmic reticulum. Also, scale bars should be added to all images of the supplementary file.

A: We agree with Reviewer #3 that a more thorough commentary on the ultrastructure in the beginning of the results section would be beneficial for the manuscript. Based on your suggestion, we acquired new data from the Spurr and Lowicryl embedded samples. We included a description of the ultrastructural preservation in Spurr and Lowicryl embeddings at the start of the results section (highlighted in green), with the introduction of a new Supplementary figure (Fig. S1) highlighting some of the features. We agree that the comparison between the Spurr and Lowicryl embedding protocols are best seen on the architecture of organelles, so the supplementary figure 1 includes details of mitochondria, endoplasmic reticulum/ Golgi and plasmodesmata for each protocol. Scale bars have been added to the supplementary figure as well as the supplementary file – EM gallery, which is now referenced altogether with supplementary figure in the results section (highlighted green).

REVIEWERS' COMMENTS

Reviewer #1 (Remarks to the Author):

The authors have addressed the issues.

Reviewer #3 (Remarks to the Author):

The authors have adequately addressed all my concerns.